# Perovskite-Based X-ray Detectors

**DOI:** 10.3390/nano13132024

**Published:** 2023-07-07

**Authors:** Chen-Fu Lin, Kuo-Wei Huang, Yen-Ting Chen, Sung-Lin Hsueh, Ming-Hsien Li, Peter Chen

**Affiliations:** 1Department of Photonics, National Cheng Kung University, Tainan 70101, Taiwan; 2Photovoltaic Technology Division, Green Energy & Environment Research Laboratories, Industrial Technology Research Institute, Tainan 71150, Taiwan; 3Department of Applied Materials and Optoelectronic Engineering, National Chi Nan University, Nantou 54561, Taiwan; 4Core Facility Center (CFC), National Cheng Kung University, Tainan 70101, Taiwan; 5Hierarchical Green-Energy Materials (Hi-GEM) Research Center, National Cheng Kung University, Tainan 70101, Taiwan; 6Program on Key Materials, Academy of Innovative Semiconductor and Sustainable Manufacturing, National Cheng Kung University, Tainan 70101, Taiwan

**Keywords:** X-ray detector, photoelectronic effect, perovskite, inorganic perovskite, two-dimensional layered perovskite, double perovskite, lead-free perovskite

## Abstract

X-ray detection has widespread applications in medical diagnosis, non-destructive industrial radiography and safety inspection, and especially, medical diagnosis realized by medical X-ray detectors is presenting an increasing demand. Perovskite materials are excellent candidates for high-energy radiation detection based on their promising material properties such as excellent carrier transport capability and high effective atomic number. In this review paper, we introduce X-ray detectors using all kinds of halide perovskite materials along with various crystal structures and discuss their device performance in detail. Single-crystal perovskite was first fabricated as an active material for X-ray detectors, having excellent performance under X-ray illumination due to its superior photoelectric properties of X-ray attenuation with μm thickness. The X-ray detector based on inorganic perovskite shows good environmental stability and high X-ray sensitivity. Owing to anisotropic carrier transport capability, two-dimensional layered perovskites with a preferred orientation parallel to the substrate can effectively suppress the dark current of the device despite poor light response to X-rays, resulting in lower sensitivity for the device. Double perovskite applied for X-ray detectors shows better attenuation of X-rays due to the introduction of high-atomic-numbered elements. Additionally, its stable crystal structure can effectively lower the dark current of X-ray detectors. Environmentally friendly lead-free perovskite exhibits potential application in X-ray detectors by virtue of its high attenuation of X-rays. In the last section, we specifically introduce the up-scaling process technology for fabricating large-area and thick perovskite films for X-ray detectors, which is critical for the commercialization and mass production of perovskite-based X-ray detectors.

## 1. Introduction

X-rays are a form of high-energy radiation that is extremely penetrating and unobservable to the naked eye. When X-rays pass through an object, it will be affected by the object and its radiation intensity will be attenuated to ionize the electron from the object. As a result of exposure to different doses and energies of X-rays, the absorber layer produces different output signals, which are then converted into observable digital images by the thin-film transistor arrays (TFT arrays) underneath the absorber layer [1]. With the development of ionizing radiation technology, X-ray detection has a wider application in medical diagnosis, non-destructive industrial radiography and safety inspection, and especially, medical diagnosis realized by medical X-ray detectors is presenting an increasing demand. Due to the low X-ray response of traditional medical X-ray detectors, high-dose X-ray exposure and prolonged exposure time are required, which have negative effects on the subject’s body. Therefore, it is necessary to develop low-cost, low-risk and high-quality X-ray detectors with high sensitivity and a low detection limit. In recent years, many researchers have proposed to replace the commonly used X-ray absorber layer of amorphous selenium (a-Se) with halide perovskite materials because of their high atomic number, long carrier lifetime, high defect tolerance, long electron–hole drift length, large bandgap and high X-ray absorption coefficient. A high X-ray absorption coefficient and other photoelectric characteristics make halide perovskite materials excellent candidates for X-ray detection.

### 1.1. Indirect-Conversion X-ray Detectors

The basic component of an indirect X-ray detector is a scintillator that converts high-energy photons, such as X-rays and gamma rays, into low-energy ultraviolet or visible light. These low-energy photons detected by an array of photoelectric diodes behind the scintillator are further converted into recognizable signals. Traditionally used materials in scintillators for indirect X-ray detection, such as CsI, NaI, Bi_4_Ge_3_O_12_, CdWO_4_ and (Lu,Y)_2_SiO_5_ [2,3,4,5,6], are well developed with some drawbacks, including complex and expensive processes, radioactive afterglow and untunable scintillation caused by uncontrollable transition energy gaps. These drawbacks could be addressed by applying the emerging perovskite as a scintillator material because of its tunable energy gap and short photoluminescence (PL) lifetime [7,8]. Since 1993, researchers have observed strong PL emission of CsPbCl_3_ with an emission lifetime in the nanosecond timescale, and it was considered a potential material for scintillators [9]. By 1995, Belsky’s team published the first research paper using perovskite as a scintillator [10]. In 2002, two-dimensional layered perovskite exhibited a strong radiative emission at room temperature [11]. To date, many research teams have developed efficient perovskite scintillators.

### 1.2. Direct-Conversion X-ray Detectors

A direct X-ray detector mainly consists of an absorber layer and a TFT array, in which the absorber layer is directly exposed to the X-ray and ionized to form different intensities of electronic signals. The electronic signals read out by the TFT array eventually show the digital image. Currently, the commercial product of a direct X-ray plate mainly employs stable a-Se due to its facile deposition directly onto large-area TFT array substrates by a low-temperature process [12]. However, due to the low atomic number of selenium, it has poor X-ray blocking and absorption ability, and the charge transport capability of selenium is also poor. It requires a large applied electrical field to drive charge carriers to the electrode to improve the carrier collection efficiency. Many research teams have successively searched for alternative materials with high X-ray absorption ability, such as mercury iodide (HgI_2_), lead iodide (PbI_2_), lead oxide (PbO) and cadmium telluride (CdTe) [13,14,15,16]. These materials have shown their potential for large-area hard X-ray (>30 keV) imaging, but these materials still have many difficulties to overcome in terms of film growth, device fabrication and operation. In recent years, emerging perovskite is a potential candidate for direct X-ray detectors due to the advantages of a low-cost solution process, excellent X-ray attenuation and absorption capabilities, high carrier mobility and long carrier lifetime. Typically, perovskite-based X-ray detectors apply two main device architectures: photoconductor- and photodiode-type structures. In the photoconductor-type structure, the intrinsic perovskite semiconductor serving as an X-ray active layer is sandwiched between two ohmic metal electrodes. When the X-ray is irradiated onto the photoconductor-type device, carriers generated by the X-ray are separated and driven by the bias voltage and then collected by the two metal electrodes. For the photodiode-type structure, which is similar to the photovoltaic devices, the perovskite active layer is in contact with at least one selective contact (n- or p-type semiconductor) to produce Schottky, p–n or p–i–n heterojunctions. The junction barrier in this device structure can facilitate the carrier’s separation under a lower bias voltage and reduce the dark current. In the past few years, various perovskite materials have been applied for direct X-ray detectors to deliver remarkable device performance with high detective sensitivity and a low detection limit [17,18].

### 1.3. Perovskite-Based X-ray Detectors

Halide perovskite, with a general formula of ABX_3_ (A-site cation: CH_3_NH_3_^+^ (methylammonium, MA^+^), HC(NH_2_)_2_^+^ (formamidinium, FA^+^) and Cs^+^; B-site cation: Sn^2+^ and Pb^2+^; X-site anion: Cl^−^, Br^−^ and I^−^), has become a promising semiconductor material for optoelectronic applications. Halide perovskite was initially employed as a light absorber in photovoltaic devices, and its photovoltaic performance quickly reached a high power conversion efficiency that is comparable to that of conventional Si- and CdTe-based solar cells [19]. The great achievement of perovskite in solar cells arouses their application in various optoelectronic devices [20]. Based on promising material properties such as excellent carrier transport properties, high effective atomic number and tunable composition of perovskite, perovskite materials are excellent candidates for high-energy radiation detection. First, in 2013, Kanatzidis’ team successfully demonstrated melt-growth CsPbBr_3_ for direct X-ray radiation detection, initiating the development of perovskite-based X-ray detectors [21]. Soon after, in 2017, Park’s team fabricated a polycrystalline MAPbI_3_ perovskite as an X-ray absorber layer that is deposited on a large-area TFT substrate using a doctor blade-coating method. The sensitivity of the fabricated device reached 1.1 × 10^4^ μC·Gy_air_^−1^·cm^−2^, which is much larger than that of common commercial X-ray detectors using a-Se [22]. In 2019, Pan’s team used the hot-pressing method to fabricate a quasi-monocrystalline structure of CsPbBr_3_, and the sensitivity of the device reached 5.5 × 10^4^ μC·Gy_air_^−1^·cm^−2^, which is still the highest sensitivity for X-ray detectors employing polycrystalline inorganic perovskites [23]. Glushkova’s team used 3D jet-printing in 2021 to fabricate a single-crystal MAPbI_3_-based X-ray detector with a record sensitivity. The sensitivity of this detector is up to 2.2 × 10^8^ μC·Gy_air_^−1^·cm^−2^, which is several orders of magnitude higher than that of the a-Se counterpart (20 μC·Gy_air_^−1^·cm^−2^) [24]. These developments demonstrate the advantages of perovskite over amorphous selenium applied in high-energy radiation detection.

## 2. Principles of High-Energy Radiation Detectors

### 2.1. Fundamentals of X-ray Detection Material

#### 2.1.1. X-ray Attenuation Ratio (ε)

When X-rays pass through matter, some interactions will occur with that matter, such as photoelectric effects, Rayleigh scattering, Compton scattering, etc. At the same time, the incident X-rays will be attenuated due to these interactions. The attenuation of X-rays can be calculated according to the Beer–Lambert law:(1)$$I=I_{0}e^{\left[-(\frac{\mu }{\rho })x\right]}$$
where I is the intensity of X-rays attenuated by the matter, I_0_ is the original intensity of incident X-ray, μ is the linear attenuation coefficient, ρ is the density of matter and x is the distance of X-ray penetration. Scientists often use the linear attenuation coefficient (μ) to compare the attenuation ability of various materials under X-ray exposure. The linear attenuation coefficient is a constant used to describe the attenuation of incident photons per unit thickness of the material, and it takes into account all possible interactions between the matter and the photon. The linear attenuation coefficient in cm^−1^ is proportional to $\frac{Z^{4}}{E_{\mathrm{ph}}{}^{3}}$, where Z is the atomic number of the material and E_ph_ is the energy of the incident photon. Sometimes the mass attenuation coefficient in cm^2^g^−1^ is also used to define the attenuation capability, which is defined as the linear attenuation coefficient per unit density of the material, yielding a value that is constant for a given element or compound. The attenuation ratio (ε) of a material with a thickness of L can be expressed as follows:(2)$$\varepsilon =1-e^{\left(-\mu L\right)}$$

Based on the previous equations, the photon absorption rate (φ), which represents the number of X-ray photons absorbed per second, can be expressed in Equation (3) [23]:(3)$$\varphi =\frac{\varepsilon \mathrm{Dm}}{E_{\mathrm{ph}}}$$
where D is the dose rate and m is the material mass.

#### 2.1.2. Ionization Energy (W)

The ionization energy (W) represents the energy required to generate a free electron–hole pair for the target material. For most semiconductor materials, the ionization energy is only related to its energy gap (Eg) and follows the empirical formula [25]:W = A ∗ Eg + B(4)
where A and B are constants. The literature indicates that the ionization energy of most perovskite materials is W = 2Eg + 1.43 eV, which is nearly an order of magnitude lower than that of amorphous selenium [23]. Under the same dose of high-energy photons, the number of free electron–hole pairs generated in perovskites is an order of magnitude higher than that of amorphous selenium. The result suggests that perovskite is an excellent candidate material as the absorber layer for a high-energy photon detector.

#### 2.1.3. Charge Collection Efficiency (CCE)

The high-energy photon incident on a material will transfer its energy to the inner shell electron of the material and excite an electron, and the kinetic energy of this photoelectron is equal to the energy of a high-energy photon minus the ionization energy of the material. This photoelectron further ionizes other low-ionization-energy electrons in the material. Theoretically, the number of electron–hole pairs produced by a high-energy photon can be calculated by the formula:(5)$$\beta =\frac{E_{\mathrm{ph}}}{W}$$
where β is the maximum number of photo-generated carriers. When the device is driven by an external electric field under illumination, the light-generated electron–hole pairs will drift to the electrode and be extracted to generate an electronic signal. The theoretically maximum photo-generated current (I_P_) can be calculated by the equation:(6)$$I_{p}=\varphi \beta e$$
where e is the quantity of electron charge. The actual number of carriers that can be finally collected by the electrodes is reduced due to carrier recombination, carrier trapping, etc. Considering the carrier loss, the modified Hecht equation [26] can be used to present the actual photogeneration current collected by the electrodes:(7)$$I=I_{0}\frac{\mu \tau V}{L^{2}}\frac{1-\exp\left(-\frac{L^{2}}{\mu \tau V}\right)}{1+\frac{L}{V}\frac{s}{\mu }}$$
where I_0_ is the saturation photocurrent, L is the material layer thickness, V is the applied bias voltage, s is the surface recombination rate, τ is the carrier lifetime and μ is the carrier mobility, and the charge collection efficiency (CCE) is defined as [27]:(8)$$\mathrm{CCE}=\frac{\mu \tau V}{L^{2}}\left[1-e^{\left(-\frac{L^{2}}{\mu \tau V}\right)}\right]$$

### 2.2. Parameters of X-ray Detectors

#### 2.2.1. Dark Current (I_dark_)

Dark current is the current generated by the device due to its environment in the absence of X-ray irradiation. The dark current in the detector belongs to the background noise signal. Excessive dark currents result in indistinguishable electronic signals, reduce the sensitivity and increase the minimum detection limit of the device when the device is exposed to X-ray irradiation.

#### 2.2.2. Sensitivity (S)

Sensitivity is a criterion to describe the ability of a direct X-ray detector to convert an incident X-ray photon into an electronic signal. Briefly, an X-ray detector with high sensitivity can produce a large electronic signal under the same exposure dose rate and enhance electronic signal identification. The sensitivity (S) of a direct X-ray detector is defined as the current density difference between the output current density with and without X-ray irradiation over the X-ray irradiation dose rate of D [28].
(9)$$S=\frac{I_{X\text{-}\mathrm{ray}}-I_{\mathrm{dark}}}{\mathrm{DA}}$$
where I_dark_ is the dark current, I_X-ray_ is the output current (I_signal_ + I_dark_) with X-ray irradiation, D is the exposure dose rate (unit in Gy·s^−1^ or R·s^−1^) and A is the sensing area. Some researchers also define the device sensitivity as [29]
(10)$$S=\frac{I_{X\text{-}\mathrm{ray}}-I_{\mathrm{dark}}}{\mathrm{DV}}$$
where V is the sensor sensing volume. The device sensitivity is affected by the X-ray attenuation, electron–hole generation, carrier extraction and photoconductivity gain.

#### 2.2.3. Limit of Detection

According to the definition of the International Union of Pure and Applied Chemistry (IUPAC) [30], the lowest detection limit is the value at which the X-ray detector can still produce a signal-to-noise ratio (SNR) = 3 at a specific exposure dose, and the corresponding exposure dose is defined as the lowest dose rate. The signal-to-noise is shown in the following formula:(11)$$\mathrm{SNR}=\frac{J_{s}}{J_{n}}$$
where J_s_ is the difference between the average photocurrent density (J_p_) and the dark current density (J_d_) and J_n_ is the noise current density, defined as the standard deviation of photocurrent density, which can be expressed as
(12)$$J_{n}=\sqrt{\frac{1}{N}\sum_{i}^{N}{\left(J_{i}-J_{p}\right)}^{2}}$$

The lower the detection limit of the X-ray detector achieved, the lower the exposure required for the device.

#### 2.2.4. Mobility-Lifetime Product (μτ)

Mobility-lifetime product is an important parameter that is used to describe the quality of an X-ray photoactive layer. A high mobility-lifetime product enhances the carrier collection efficiency of the detector and improves the device’s performance. Theoretically, the mobility-lifetime product value is correlated with the CCE by the Hecht equation, as presented in Equation (8) [26].

#### 2.2.5. Response Time

Response time is used to quantify the sensing speed of a detector. The general definition of response time is usually expressed in the form of rising time and falling time. Rising time is the time of photocurrent required to rise from 10% to 90% of the saturated photocurrent when the X-ray detector is exposed to X-ray irradiation, while the falling time is the time of photocurrent required to fall from 90% to 10% of the saturated photocurrent when the X-ray irradiation is off. The falling time is usually longer than the rising time because of the trap or defect states in the material. The ideal X-ray detectors should have a short response time, which not only reduces the exposure time to X-rays but also facilitates their use in X-ray imaging, such as fluoroscopy. The response time of perovskite-based X-ray detectors varies from a few milliseconds to sub-milliseconds, which makes it difficult to apply them in X-ray image acquisition at high frame rates.

## 3. Classification of Perovskite-Based X-ray Detectors

Single crystals (SCs), wafers and film types of halide perovskites are developed for X-ray detection. The perovskite single crystals and wafers have few grain boundaries and a high μτ value. Film-type perovskite could be directly deposited onto a flexible substrate with a large-size area for specific X-ray detection. Therefore, we highlight some advanced works in halide perovskite-based X-ray detectors using perovskite SCs, wafers and films.

### 3.1. Three-Dimensional (3D) ABX_3_ Structure

#### 3.1.1. Organic/Inorganic 3D ABX_3_ Structure

##### Single Crystals (SCs)

An X-ray detector using single-crystal perovskite usually has high X-ray sensitivity, high bulk resistivity and a low trap density due to its good material quality with smooth morphology. The inverse temperature crystallization (ITC) method is a traditional method for fabricating single crystals. The ITC process can shorten the reaction time and prepare high-quality perovskite single crystals. Many studies have demonstrated the fabrication of organic–inorganic hybrid perovskite single crystals, such as MAPbI_3_ and MAPbBr_3_. Among them, Song et al. [31] used the surface passivation strategy with methylammonium iodide (MAI) to remarkably increase the ion migration activation energy of MAPbI_3_ SCs (as seen in Figure 1a) and effectively stabilize the dark current of a coplanar-structure device. Under a large electric field of 100 V mm^−1^, the as-fabricated device achieved a record-high sensitivity above 700,000 μC·Gy_air_^−1^·cm^−2^ (X-ray irradiation with energy up to 50 keV). Wang et al. [32] used the continuous mass transport process (CMTP) with steady self-supply to improve the carrier mobility of single-crystal MAPbI_3_ perovskite. The single-crystal MAPbI_3_ perovskite prepared by CMTP exhibited a low trap density of 4.5 × 10^9^ cm^−3^, a high carrier mobility of 150.2 cm^2^·V^−1^·s^−1^ and a high mobility–lifetime product of 1.6 × 10^−3^ cm^2^·V^−1^. Under soft X-ray irradiation with a Cu Kα peak energy at 8 kV, the X-ray detector using MAPbI_3_ single crystal prepared by CMTP delivered a superior X-ray response than that prepared by ITC. Huang et al. [33] employed A-site cation engineering to fabricate alloyed DMAMAPbI_3_ (DMA: dimethylammonium) and GAMAPbI_3_ (GA: guanidinium) SCs via ITC. Alloying large-sized A-site cations of DMA^+^ or GA^+^ can improve the charge collection efficiency presumably due to the increased defect formation energy and the decreased electron–phonon coupling strength. The X-ray detectors based on GAMAPbI_3_ exhibited the best performance with a sensitivity of 2.31 × 10^4^ µC·Gy^−1^·cm^−2^ and a detection limit of 16.9 nGy_air_·s^−1^. Ye et al. [34] used seed dissolution–regrowth to improve the crystal quality of cuboid MAPbI_3_. The MAPbI_3_-based X-ray detector delivered a sensitivity of 968.9 µC·Gy^−1^·cm^−2^ under −1 V bias. Later, Geng et al. [35] controlled the temperature gradient (TG) during the synthesis of MAPbI_3_ SCs to effectively reduce the trap density and improve the crystal quality (refer to Figure 1b). Such an improvement led to the as-fabricated X-ray detector having a boosted sensitivity of 1471.7 μC·Gy_air_^−1^·cm^−2^ under a low electric field of 3.3 V·mm^−1^. Shrestha et al. [36] reported a mechanical sintering process to fabricate wafer-sized perovskite with polycrystalline MAPbI_3_ that was several hundred micrometers thick. The fabricated device showed a sensitivity of 2527 µC·Gy_air_^−1^·cm^−2^ under 70 kV_p_ X-ray irradiation and a high ambipolar mobility–lifetime product of 2 × 10^−4^ cm^2^·V^−1^.

The MAPbBr_3_ perovskite exhibits a higher stability than the MAPbI_3_ one and a wider absorption range for visible light than the MAPbCl_3_ one. Geng et al. [37] synthesized a high-quality MAPbBr_3_ SC with a high mobility–lifetime product of 4.1 × 10^−2^ cm^2^·V^−1^ by the ITC process, and its application for the X-ray detector exhibited a high sensitivity up to 259.9 μC·Gy_air_^−1^·cm^−2^ under the X-ray irradiation of 39 keV. Xu et al. [38] fabricated a reliable and sensitive MAPbBr_3_-based X-ray detector with a high on–off photocurrent ratio and a fast response based on the Au-MAPbBr_3_–Al sandwich structure (as seen in Figure 1c). The Schottky barrier built at the MAPbBr_3_–Al heterojunction effectively suppressed the leakage current and enhanced the charge collection capability. Eventually, the corresponding device exhibited a high sensitivity of 359 μC·Gy_air_^–1^·cm^–2^ and a fast response time of 76.2 ± 2.5 μs at room temperature. To improve the sensitivity of the X-ray detector, Pan et al. [39] proposed a perovskite–perovskite heterojunction formed by the epitaxial growth of MAPbBr_3_ SCs on Bi^3+^-doped MAPbCl_3_ SCs. The perovskite–perovskite heterojunction shows a relatively low trap density and enhanced built-in potential. The resultant X-ray detector delivered a fast response time of 4.89 μs and a high sensitivity of 1.72 × 10^3^ μC·Gy_air_^−1^·cm^−2^ for 50 kVp X-ray photon illumination under a reverse electric field of 31.5 V·mm^−1^. Fan et al. [40] showed an effective method to fabricate mixed-cation MA_x_Cs_1−x_PbBr_3_ SCs with the assistance of antisolvent and successfully used the mixed-cation perovskite SCs to produce a highly sensitive X-ray detector with a symmetrical sandwich structure. Their bandgap can be tuned from 2.25 to 2.16 eV by increasing the MA composition, and adding MA at the A site results in significantly improved electronic properties of the mixed-cation perovskite involving lower trap density, higher mobility and higher conductivity. The X-ray detector employing mixed-cation MA_x_Cs_1−x_PbBr_3_ perovskite achieved a remarkable sensitivity of up to 2017 μC·Gy_air_^−1^·cm^−2^ and a detection limit of 1.2 × 10^3^ nGy_air_·s^−1^ under an applied voltage of 1 V.

Although thermal evaporation is an expensive process for fabricating perovskite SC, it is feasible to produce higher-quality SCs with a larger size than those prepared by the solution process. Liu et al. [41] fabricated stable, inch-sized and multi-component perovskite (FAMACs) SCs, in which the MA^+^ and cesium (Cs^+^) cations and the bromine (Br^−^) anion were mixed into the FAPbI_3_ lattice by thermal evaporation (refer to Figure 1d). Their application for the X-ray detector showed a high sensitivity of (3.5 ± 0.2) × 10^6^ μC·Gy_air_^−1^·cm^−2^ and a detection limit of 42 nGy_air_·s^−1^ under 40 keV X-ray radiation. This work demonstrated the exclusive ability to recognize the images of steel objects in a closed opaque black plastic box by X-ray detection.

**Figure 1 nanomaterials-13-02024-f001:**
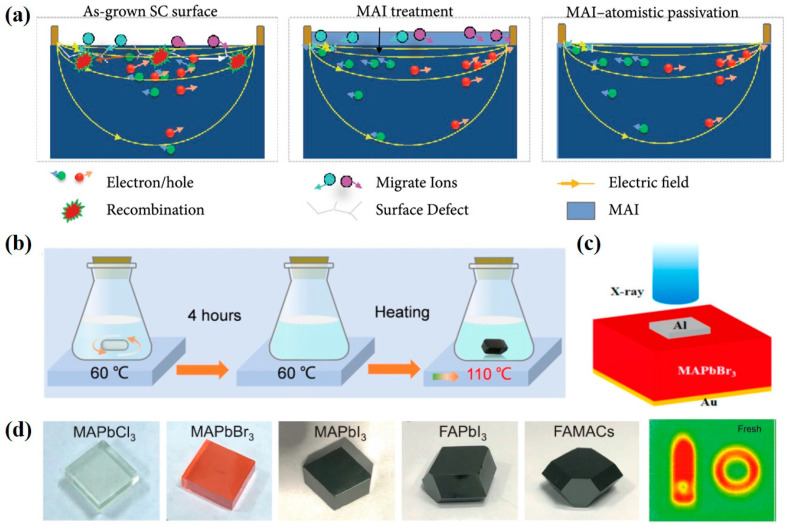
Application of organic–inorganic hybrid perovskite SCs in X-ray detectors. (**a**) Schematic diagram of the working principle for the X-ray detector with different surface treatments [31]. (**b**) Schematic diagram of the crystallization process for the MAPbI_3_ SCs with small and large temperature gradients [35]. (**c**) Schematic diagram of a device with Au/MAPbBr_3_ SCs/Al architecture [38]. (**d**) Different perovskite SCs fabricated by thermal evaporation and X-ray images of a closed black plastic box measured by the FAMACs-SC-based X-ray detector [41].

##### Thin Films

There are only a few studies about the fabrication of perovskite thin film applied for X-ray detection, and the main reason is the lower X-ray attenuation ability of perovskite thin film than that of perovskite SCs. The general method for preparing perovskite thin film is the spin-coating process. Yang et al. [32] reported that a 700 nm MAPbI_3_ thin film can only survive under a high bias of 5 V for 15 min. However, the X-ray detector based on the MAPbI_3_ thin film shows a normal sensitivity of 2.48 × 10^−2^ μC·Gy_air_^−1^·cm^−2^. Basiricò et al. [42] reported triple-cation perovskite (Cs_0.05_FA_0.79_MA_0.16_Pb(I_0.8_ Br_0.2_)_3_) thin films applied in an X-ray detector, which presented a sensitivity 3.7 ± 0.1 µC·Gy^−1^·cm^−2^ without bias. The good device performance is ascribed to two major factors: (i) high mobility-lifetime (µτ) product in the triple-cation perovskite due to a low degree of disorder, and (ii) efficient electron and hole transport due to ambipolar charge transport in the triple-cation perovskite. Mixing n-type iodide with p-type bromide in the perovskites provides high electron and hole mobilities that grant an efficient charge collection and a fast detection response for the device. In addition, there are also some novel production methods to fabricate the perovskite. Possanzini et al. [43] reported fully textile perovskite-based direct X-ray detectors, in which the photoactive layer was constituted by a silk satin fabric functionalized with MAPbBr_3_. Sensitivity values up to 12.2 ± 0.6 µC·Gy^−1^·cm^−2^ and a limit of detection down to 3 × 10^3^ nGy_air_·s^−1^ were achieved.

#### 3.1.2. Inorganic 3D ABX_3_ Structure

##### Single Crystals (SCs)

CsPbBr_3_ perovskite has high X-ray attenuation ability, a highly effective atomic number, high resistance and better long-term stability than the organic counterpart. These advantages attracted many researchers’ attention to boost the development of CsPbBr_3_-based X-ray detectors. Fan et al. [44] applied a hole extraction layer of MoO_3_ on the CsPbBr_3_ photoactive layer to increase the hole carrier collection in the X-ray detector and improve its signal current under the X-ray radiation. The sensitivity of X-ray detectors after covering the MoO_3_ layer can reach up to 2552 μC·Gy_air_^−1^·cm^−2^ under an electric field of 45 V·cm^−1^. Zhang et al. [45] prepared CsPbBr_3_ perovskite by a low-temperature solution method, which exhibited high transmittance and large mobility-lifetime products for the X-ray detector with an asymmetric electrode configuration, and the ion migration in the device was effectively suppressed even under a high voltage. The device sensitivity achieved 1256 μC·Gy_air_^−1^·cm^−2^ for 80 kVp X-ray radiation under an electric field of 20 V·mm^−1^. Peng et al. [46] developed a low-temperature crystallization strategy to grow CsPbBr_3_ SCs in water. The hole and electron mobilities of CsPbBr_3_ SCs reached 128 and 160 cm^2^·V^−1^·s^−1^, respectively. Finally, the device using CsPbBr_3_ SCs showed a high X-ray sensitivity of 4086 μC·Gy_air_^−1^·cm^−2^.

##### Other Structures

Other structures of CsPbBr_3_, such as nanocrystals [47], microcrystals [48,49], quasi-monocrystals [23] and thin films [50], have been developed for X-ray detectors. Gou et al. [48] achieved a self-powered X-ray detector based on a microcrystalline CsPbBr_3_ thick film by the solution process. A sensitivity of 470 µC·Gy_air_^−1^·cm^−2^ was obtained for the X-ray photodetector under a low dose rate 0.053 µGy_air_·s^−1^ without bias. Matt et al. [49] employed the melting process to fabricate CsPbBr_3_ microcrystals that showed high crystallinity and chemical purity. The film featured a resistance of 8.5 GΩ cm and a hole mobility of 18 cm^2^·V^−1^·s^−1^. The corresponding device showed a sensitivity of 1450 µC·Gy_air_^−1^·cm^−2^ under an electric field of 1.2 × 10^4^ V·cm^−1^ and a detection limit below µGy_air_·s^−1^. Pan et al. [23] used a hot-pressing method to produce thick quasi-monocrystalline CsPbBr_3_ films with uniform crystalline orientations. They mentioned that the high crystalline quality of the CsPbBr_3_ films and the self-formed shallow bromide vacancy defects during the high-temperature process result in a large µτ product. The X-ray detector using thick quasi-monocrystalline CsPbBr_3_ films shows a sensitivity of 55,684 µC·Gy_air_^−1^·cm^−2^, surpassing the sensitivity of all other perovskite-based X-ray detectors. A recorded sensitivity was achieved for the perovskite-based X-ray detector made by the all-vacuum deposition process. Lai et al. [50] demonstrated a perovskite-based X-ray detector with a p–i–n heterojunction, in which the Cs-based perovskite active layer was fabricated by vacuum deposition. The self-powered X-ray detector showed an efficient charge collection, an exceptionally high X-ray sensitivity of 1.2 C·Gy_air_^–1^·cm^–3^ and a low detection limit of 25.69 nGy_air_·s^–1^ under zero-bias conditions. Moreover, the volume sensitivity (C·Gy_air_^–1^·cm^–3^) was only one-fifth of that of the vacuum-deposited CsPbI_2_Br devices.

### 3.2. Low-Dimensional Perovskite Materials

Low-dimensional perovskite materials are mostly used in light sensors and LEDs [51]. Low-dimensional perovskite material is characterized by a layered crystal structure, in which the corner-sharing BX_6_ octahedra are separated between two spacers of large-sized R-NH_3_^+^ cations. The structural formula of low-dimensional layered perovskite can be presented as (R-NH_3_)_2_A_n−1_B_n_X_3n+1_ (Ruddlesden–Popper phase). When n is equal to 1, a basic two-dimensional layered perovskite with a formula of (R-NH_3_)_2_BX_4_ is obtained. By controlling the layer number n of BX_6_ octahedra, a variety of low-dimensional perovskite materials are formed. When the layer number of BX_6_ octahedra is close to infinity, the two-dimensional layered perovskite can convert into a three-dimensional perovskite. In the two-dimensional layered perovskite, the carrier transport is limited within the layered BX_6_ octahedra and leads to anisotropic carrier transport because of a strong dielectric difference and a weak bonding between the large-sized R-NH_3_^+^ and the inorganic BX_6_ octahedra. A low dark current of the X-ray detector based on the low-dimensional perovskite is achieved due to suppressed ion migration and large E_g_ [52].

#### 3.2.1. Two-Dimensional (2D) Perovskite Materials

The architecture and material properties of two-dimensional (2D) perovskite materials as well as their advantages were briefly introduced. Next, the recent research on the application of two-dimensional materials in X-ray sensing devices is summarized.

##### Single Crystal

Li et al. [53] used a 2D perovskite of (F-PEA)_2_PbI_4_ (F-PEA: fluorophenethyl) material as the active layer for the hard X-ray detector. Adding the fluorine atoms in the large-sized cation can strengthen the interaction between the fluorine atom and the benzene ring, which improves the stability, effectively suppresses the ion migration and reduces the dark current of the X-ray detector. Figure 2a is the structural architecture of the device. The device sensitivity under 120 kV_p_ hard X-ray irradiation is 3402 μC·Gy^−1^_air_·cm^−2^ and its detection limit is 23 nGy_air_·s^−1^, as shown in Figure 2b and Figure 2c, respectively. Qian et al. [54] synthesized a 2D (PMA)_2_PbI_4_ single crystal and used the ion implantation to implant copper ions into the 2D perovskite (refer to Figure 2d). The quantum confinement effect of 2D layered perovskite can effectively suppress the dark current of the X-ray detector. The device sensitivity under the X-ray irradiation is 283 μC·Gy_air_^−1^·cm^−2^ and its detection limit is 2.13 μGy_air_·s^−1^, as shown in Figure 2e and Figure 2f, respectively.

Li et al. [55] fabricated a single crystal of (F-PEA)_3_BiI_6_ by pressing disordered two-dimensional perovskite powders into thick films with ordered single-crystal structures under high pressure. Figure 2g is the process of pressure tableting. Strong bonding energy between bismuth ions (Bi^3+^) and iodine ions (I^−^) can effectively prevent ion migration and reduce the dark current of the X-ray detector. The device detection limit is 30 nGy_air_·s^−1^ and the electrical detection of different crystal orientations is compared in Figure 2h,i.

##### Film

Ledee et al. [51] deposited 2D layered perovskite of PEA_2_PbBr_4_ (PEA = C_6_H_5_C_2_H_4_NH^3+^) on a PET flexible substrate coated with interdigitated electrodes, and such a device structure works as a photoconductor, as shown in Figure 3a. The channel width and length of finger electrodes were designed as 7.56 mm and 20 µm, respectively, to assist the layered arrangement of 2D material and effectively collect the charge carriers. As shown in Figure 3b, when the dose rate increases, the sensitivity of X-ray detectors decreases. The sensitivity is calculated as the first derivative of photocurrent with respect to the dose rate and it is normalized by the total pixel area of 0.63 mm^2^. The device under X-ray irradiation of 150 kVp presented a sensitivity of 806 μC·Gy_air_^−1^·cm^−2^. Figure 3c shows the SNR as a function of dose rate. When the SNR of the X-ray detector is equal to 3, the corresponding detection limit reaches a low value of 42 nGy_air_·s^−1^.

Compared with the single crystal, the perovskite film has more defects, which results in increased dark current in the X-ray detector and thereby decreased device sensitivity. Due to the large-sized cation in the 2D perovskite, the ion migration can be inhibited to reduce the dark current of the device. However, the 2D perovskite has a poor light response to X-rays due to anisotropic charge transport. As a result, the device applying 2D perovskite delivers an inferior sensitivity compared to the 3D counterpart.

#### 3.2.2. Quasi-2D Perovskite Materials

Ji et al. [56] produced a quasi-2D BA_2_EA_2_Pb_3_Br_10_ (BA: C_4_H_9_NH_3_; EA: C_2_H_5_NH_3_) perovskite as an X-ray absorber for the X-ray detector, and they utilized spontaneous ferroelectric polarization (Ps) to effectively separate the photoinduced carriers and facilitate their transport for better X-ray sensitivity, as shown in Figure 4a,b. Tsai et al. [57] used the 2D layered perovskite of (BA)_2_(MA)_2_Pb_3_I_10_ as the active layer material in an X-ray detector for converting X-rays into charges, and the device sensitivity reached 0.276 C·Gy_air_^−1^·cm^−3^, as shown in Figure 4c,d. In addition, Tsai et al. [58] also introduced *n*-butylamine iodide in the methylammonium lead iodide precursor to prepare a quasi-2D layered perovskite, as shown in Figure 4e. The corresponding device showed a stable performance for a detection time of more than 15 h, and the X-ray sensitivity of the device was 1214 µC·Gy_air_^−1^·cm^−2^, as shown in Figure 4f.

#### 3.2.3. One-Dimensional (1D) Perovskite Materials

Zhang et al. [59] synthesized a 1D inorganic perovskite of CsPbI_3_ as an absorbing material for the conversion of X-ray photons into charge carriers, and the CsPbI_3_-based device had a high bulk resistivity of 7.4 × 10^9^ Ω·cm, a large mobility–lifetime product of 3.63 × 10^−3^ cm^2^·V^−1^ and X-ray sensitivity of 2.37 mC·Gy^−1^·cm^−2^.

#### 3.2.4. Zero-Dimensional (0D) Perovskite Materials

Unlike the traditional 3D perovskite structures with corner-sharing BX_6_ octahedra, deficient perovskite has isolated BX_6_ octahedra, which results in a zero-dimensional (0D) perovskite structure. Xu et al. [60] used electrostatic-assisted spray coating to fabricate the all-inorganic 0D perovskite material of Cs_2_TeI_6_ with a large area for the X-ray detector. Figure 5a is the schematic illustration of the spraying process for preparing the Cs_2_TeI_6_ perovskite, and the device structure of the X-ray detector is composed of FTO/TiO_2_/Cs_2_TeI_6_/PTAA/Au (refer to the inset in Figure 5b). Under X-ray irradiation of 40 kVp and an electric field of 250 V·cm^−1^, the device sensitivity was 19.2 μC·Gy_air_^−1^cm^−2^. Figure 5b compares the X-ray attenuation efficiency of different active materials, including CdTe, a-Se and Cs_2_TeI_6_, with different thicknesses, indicating that the Cs_2_TeI_6_ perovskite exhibits a comparable X-ray attenuation efficiency to CdTe. Xu et al. [61] used the 0D single crystal of Cs_4_PbI_6_ perovskite to make an X-ray detection device, and its mobility–lifetime product (μτ) reached 9.7 × 10^−4^ cm^2^·V^−1^. Figure 5c shows the J–V curves of the fabricated device with different exposure dose rates and the resultant sensitivity of the device was 451.49 μC·Gy_air_^−1^·cm^−2^ at an operating bias of 30 V, as seen in Figure 5d.

### 3.3. A_2_B_2_X_6_ Double-Perovskite Materials

Inorganic Cs_2_AgBiBr_6_ double perovskite is the most popular material in the field of X-ray detection, and its advantages include elements with a large Z number, such as Ag and Bi, which have a high X-ray attenuation ability. When a large electric field is applied to the Cs_2_AgBiBr_6_-based X-ray detector, the Cs_2_AgBiBr_6_ double perovskite is durable. Moreover, the ion migration in the X-ray detector is inhibited to lower the dark current.

Steele et al. [62] used Cs_2_AgBiBr_6_ double perovskite for X-ray detection. The device architecture was composed of Au/Cs_2_AgBiBr_6_/Au stacking structure and its sensitivity to X-ray irradiation reached 105 μC·Gy_air_^−1^·cm^−2^, as shown in Figure 6a. In the research demonstrated by Pan et al. [63], they reduced the disordered arrangement of Ag^+^ and Bi^3+^ ions in the Cs_2_AgBiBr_6_ double perovskite through the control of thermal annealing and surface treatment, which increases the bulk resistivity of perovskite and reduces the dark current. It is indicated that this Cs_2_AgBiBr_6_ SC is a p-type semiconductor to facilitate hole transport.

Yuan et al. [64] proposed that phenethylamine bromide (PEABr) can strengthen the molecular arrangement in the Cs_2_AgBiBr_6_ perovskite and were able to in situ tune the order–disorder phase transition in the Cs_2_AgBiBr_6_ single crystals. Additionally, the improvement of ordering extent can effectively reduce the defect density of perovskite material and improve carrier mobility. These PEA-Cs_2_AgBiBr_6_ crystals were utilized in the application of X-ray detection. The working mechanism is shown in Figure 6b. The PEA-Cs_2_AgBiBr_6_-based device showed an enhanced X-ray response due to the improvement of carrier mobility, as shown in Figure 6c, and the sensitivity of the PEA-Cs_2_AgBiBr_6_-based X-ray detector was 288.8 µC·Gy_air_^−1^·cm^−2^.

### 3.4. A_3_B_2_X_9_ Lead-Free Perovskite Materials

Lead-free perovskite with a vacancy-ordered structure of A_3_B_2_X_9_ (A: Cs^+^, Rb^+^, NH_4_^+^; M: Bi^3+^, Sb^3+^; X: Br^−^, I^−^) is a potential candidate for X-ray detectors due to its eco-friendly composition and high attenuation coefficient to X-ray irradiation. This perovskite has an equivalent composition of AB_2/3_X_3_ that indicates that one in three octahedral B^3+^-sites is occupied by a vacancy for maintaining charge neutrality. These materials exhibit a layered structure as 2D layered perovskite derivatives. This perovskite contains heavy elements with a large Z number to deliver a high attenuation capability to X-ray irradiation based on the relationship α ∝ Z^4^/E_ph_^3^. The application of vacancy-ordered perovskite for X-ray detectors is introduced in this section.

Zhuang et al. [65] used a 2D layered perovskite single crystal of (NH_4_)_3_Bi_2_I_9_ for the X-ray detector. Due to its inherent 2D layered structure, ion migration in the (NH_4_)_3_Bi_2_I_9_ perovskite can be suppressed. The as-fabricated device presented a low detection limit of 55 nGy_air_·s^−1^ and exhibited a good X-ray attenuation performance due to the presence of the heavy element Bi. Dong et al. [66] applied the MA_3_Bi_2_I_9_ perovskite for the direct-conversion X-ray detector. They used the blade-coating method to deposit the perovskite film onto the ITO substrate and added the MACl additive to slow down the crystal growth rate to reduce pinholes in the blade-coated perovskite film. The X-ray sensitivity and the detection limit of the device were 100.16 μC·Gy_air_^−1^·cm^−2^ and 98.4 nGy_air_·s^−1^, respectively. Li et al. [67] synthesized a single-crystal, low-dimensional FA_3_Bi_2_I_9_ perovskite material as the X-ray absorber for an X-ray detector. Figure 7a shows the solution-processed method for the fabrication of the FA_3_Bi_2_I_9_ single crystal. The device had a high bulk resistivity of 7.8 × 1010 Ω·cm, and the X-ray detection limit and the sensitivity of the device were 200 nGy_air_·s^−1^ and 598.1 μC·Gy_air_^−1^·cm^−2^, respectively. Since the stability of the device is severely affected by the organic cations, Zhang et al. [68] synthesized an all-inorganic low-dimensional perovskite of Cs_3_Bi_2_I_9_ single crystal as the active layer for X-ray sensing. The Cs_3_Bi_2_I_9_ single crystal was formed by the solution method and then made into the X-ray sensing device, as shown in Figure 7b. The sensitivity and the detection limit of the device for X-ray sensing reached 1652.3 μC·Gy_air_^−1^·cm^−2^ and 130 nGy_air_·s^−1^, respectively.

According to the X-ray absorption mechanism, the active layer with heavy elements has high absorption of X-ray irradiation. Therefore, X-ray detectors using perovskite materials with heavy metal elements of bismuth have good sensitivity to X-ray irradiation. Compared with the organic–inorganic hybrid perovskites, the inorganic perovskites present superior resistance to moisture, so the inorganic Cs_3_Bi_2_I_9_ is a suitable material for X-ray sensing.

### 3.5. Large-Area Perovskite X-ray Detectors

To make the perovskite-based X-ray detector able to directly take 2D images, a large area of perovskite layers, greater than 100 cm^2^, is favorable for the integration on the commercial thin-film transistor array; therefore, a scalable deposition process for the perovskite active layer in the X-ray detector is necessary. In this section, we summarize the large-area deposition process of perovskite active layers for the manufacture of perovskite-based X-ray detectors and their performance under X-ray irradiation.

Many large-area deposition processes of perovskite active layers, such as inkjet printing, doctor blade coating, spray coating, etc., have been demonstrated. Mescher et al. [69] reported a direct-conversion X-ray detector, in which the perovskite film was inkjet-printed onto flexible substrates, as shown in Figure 8a. The as-fabricated device achieved a sensitivity of 59.9 µC·Gy_air_^−1^·cm^−2^ at a low external voltage (0.1 V) with 70 kV_p_ X-ray irradiation. Liu et al. [70] demonstrated an X-ray detector via a low-cost ink-printed coating of CsPbBr_3_ quantum dots on a flexible substrate, as shown in Figure 8b. A sensitivity of 1450 µC·Gy_air_^−1^·cm^−2^ for the X-ray detector was obtained when the device was exposed to a low exposure dose rate of about 17.2 µGy_air_·s^–1^ under a bias voltage of 0.1 V. Li et al. [71] demonstrated a hybrid X-ray detector with Cs_2_AgBiBr_6_ and (C_38_H_34_P_2_)MnBr_4_ scintillators synthesized by the tablet-pressing method, as shown in Figure 8c. The addition of a (C_38_H_34_P_2_)MnBr_4_ scintillator in the X-ray detector could suppress the ion migration of Cs_2_AgBiBr_6_ perovskite to reduce the dark current and increase the carrier collection efficiency. The final device achieved a sensitivity of 114 µC·Gy_air_^−1^·cm^−2^ and a low detection limit of 200 nGy_air_·s^–1^ under a hard X-ray exposure of 120 kV_p_. Ciavatti et al. [72] introduced a bar-coated perovskite for the direct-conversion X-ray detector with a sensitivity of 494 µC·Gy_air_^−1^·cm^−2^ at 150 kV_p_ X-ray irradiation under a low operating voltage of less than 4 V, as shown in Figure 8d. Guo et al. [73] provided a facile mobile-platform-assisted electrospray method for the preparation of perovskite films with a large area of 100 cm^2^ on flexible substrates. By adjusting the spraying parameters, it is possible to control the growth orientation of Cs_2_TeI_6_ perovskite. The Cs_2_TeI_6_-based X-ray detector, in which the perovskite shows a preferred crystalline orientation of (222) facet, achieves a resistivity of 1.9 × 10^11^ Ω·cm and a sensitivity of 226.8 µC·Gy_air_^−1^·cm^−2^, as shown in Figure 8e. Kim et al. [22] used doctor blade coating to deposit a polycrystalline MAPbI_3_ thick film onto large-area thin-film transistor (TFT) substrates. This is the first work to directly deposit the MAPbI_3_ perovskite on a large-area TFT substrate. The device achieves a sensitivity of up to 1.1× 10^4^ μC·Gy_air_^−1^·cm^−2^ under irradiation from a 100 kV_p_ X-ray source, which is higher than that of current X-ray detectors using amorphous selenium or thallium iodide. They adjusted the charge injection interface to reduce the dark current by overlaying two additional polymer/perovskite composites on the upper and lower interfaces of 830 μm thick MAPbI_3_. Yang et al. [74] proposed an aerosol–liquid–solid (ALS) process to prepare mixed halide perovskite thick films for direct-conversion X-ray detectors. The ALS process can produce high-quality perovskite films of different compositions on conductive glass substrates or TFT substrates and facilitate the vertical growth of perovskite grains without grain boundaries. Such a columnar perovskite grain significantly improves the carrier transport across the perovskite film. The X-ray detector with a CsPbI_2_Br thick film achieves a sensitivity of 1.48 × 10^5^ μC·Gy_air_^−1^·cm^−2^ and a low detection limit of 280 nGy_air_·s^−1^, as shown in Figure 8f. The authors further demonstrated the high-resolution imaging capability, good stability, and strong resistance to X-ray radiation damage for the CsPbI_2_Br-based X-ray detector.

## 4. Conclusions and Outlook

In recent years, many researchers have proposed to replace the commonly used a-Se with perovskite material as the absorber layer for X-ray detectors because of its excellent X-ray attenuation ratio, long carrier lifetime, high defect tolerance and long electron–hole drift length. Compared to an a-Se absorber layer, using emerging perovskite materials in an X-ray detector can provide better X-ray detection performance at a lower cost and with less environmental pollution. At the early stage in the development of perovskite-based X-ray detectors, single-crystal perovskite was used to replace a-Se in X-ray detectors and showed good performance in response to X-ray irradiation due to its high crystalline quality, smooth morphology, high bulk resistivity, low trap density and sufficient thickness to attenuate X-rays. Polycrystalline perovskite-based X-ray detectors show poorer performance than their single-crystalline counterparts because of the increased grain boundary. However, their fabrication methods have the potential for the mass production of thick perovskite films for large-area X-ray detectors. Organic–inorganic hybrid perovskite materials, such as MAPbI_3_ and MAPbBr_3_, are commonly used as absorbers for X-ray detectors, and their application in X-ray detectors shows good performance under X-ray irradiation. However, their environmental stability is a major challenge due to the rapid degradation of perovskite upon exposure to humidity, heat, light and oxygen. All inorganic perovskite materials, such as CsPbBr_3_, show better environmental stability than organic–inorganic perovskite, and the X-ray detectors with inorganic perovskite still have high X-ray sensitivity. The introduction of large-sized cations in the 3D perovskite separates the 3D perovskite into the 2D layered perovskite. The anisotropic carrier transport in the 2D layered perovskite can inhibit ion migration to reduce the dark current of the device but decrease the device’s performance in response to X-ray irradiation. As a result, the sensitivity of X-ray detectors using 2D perovskite is generally lower than that of their 3D perovskite counterparts. Quasi-2D perovskite material (*n* > 1) such as BA_2_EA_2_Pb_3_Br_10_ has a better response to X-rays than the pure 2D materials (*n* = 1) and still has good stability. After a long period of measurement, the X-ray detectors based on quasi-2D perovskite still maintain a very stable performance. The application of a 1D inorganic perovskite nanorod, such as CsPbI_3_, in the X-ray detector shows a high X-ray sensitivity owing to its high resistivity and large carrier mobility–lifetime product. The 0D all-inorganic perovskite material of Cs_2_TeI_6_ is demonstrated as a potential candidate for the X-ray detector by virtue of its high X-ray sensitivity and environmental stability. Double-perovskite materials, such as Cs_2_AgBiBr_6_, contain the large Z-numbered elements of Ag and Bi, which have a better ability to attenuate X-ray irradiation. Its stable crystal structure can effectively inhibit ion migration in the device under a bias voltage; therefore, the corresponding device has a low dark current. Lead-free perovskite materials with a composition formula of A_3_B_2_X_9_ (e.g., MA_3_Bi_2_I_9_ and Cs_3_Bi_2_I_9_), in which the Pb atoms are replaced with larger Z-numbered Bi atoms, show the advantages of high attenuation of X-rays and eco-friendly properties. These properties make such materials promising as active layers in X-ray sensing devices. Furthermore, all inorganic lead-free perovskite materials exhibit high resistance to moisture, so they are suitable candidates for X-ray detectors. The key device parameters of perovskite-based X-ray detectors are summarized in Table 1.

To meet the commercial requirements for flat panel X-ray detectors, direct deposition of a perovskite layer with an area larger than 100 cm^2^ on the thin-film transistor array is necessary. Large-area fabrication methods for preparing the thick perovskite film, such as inkjet printing, doctor blade coating, spray coating and the ALS process, are utilized to fabricate large-area perovskite-based X-ray detectors. The large-area fabrication methods with a low-temperature process can further deposit a perovskite thick film onto the flexible substrates or TFT modules. The improvement of perovskite film quality, film uniformity and composition uniformity are the major challenges for large-area perovskite film. The as-prepared perovskite thick film by these processes is a polycrystalline film that has abundant grain boundaries and defects to trap the photogenerated carriers and further reduce the X-ray detector performance. By optimizing the experimental parameters and precisely controlling the environment during the deposition of perovskite film, a large-area perovskite thick film with large grain size, uniform film morphology and composition and fewer defects can be obtained. In conclusion, it is worth devoting more efforts toward developing facile, low-cost and eco-friendly fabrication methods for the preparation of large-area, durable, uniform, highly sensitive, low-detection-limit and low-toxicity perovskite films for X-ray detection.

## Figures and Tables

**Figure 2 nanomaterials-13-02024-f002:**
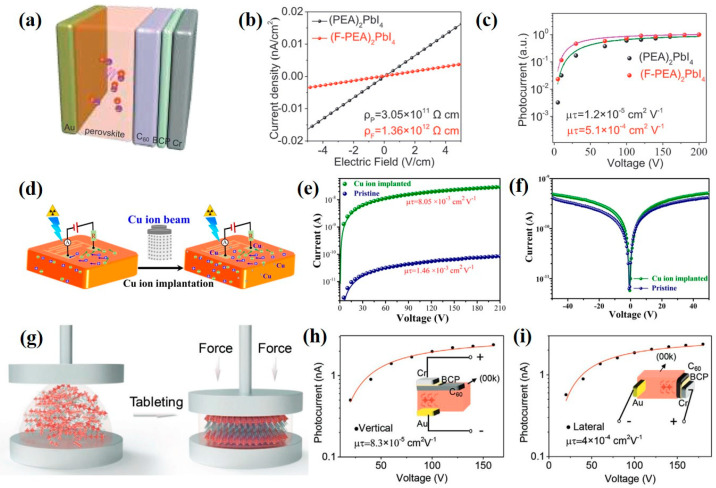
(**a**) Device structure, (**b**) dark current density and (**c**) photocurrent of self-powered X-ray detector based on 2D (F-PEA)_2_PbI_4_ perovskite [53]. (**d**) Schematic diagram of Cu ion implantation on the (PMA)_2_PbI_4_-based X-ray detector. (**e**) Sensitivity and (**f**) detection limit of (PMA)_2_PbI_4_-based X-ray detector after ion implantation [54]. (**g**) Schematic diagram of tableting process for the (F-PEA)_3_BiI_6_ SCs. Photocurrent of X-ray detector using (F-PEA)_3_BiI_6_ single crystal under (**h**) vertical and (**i**) horizontal electric fields [55].

**Figure 3 nanomaterials-13-02024-f003:**
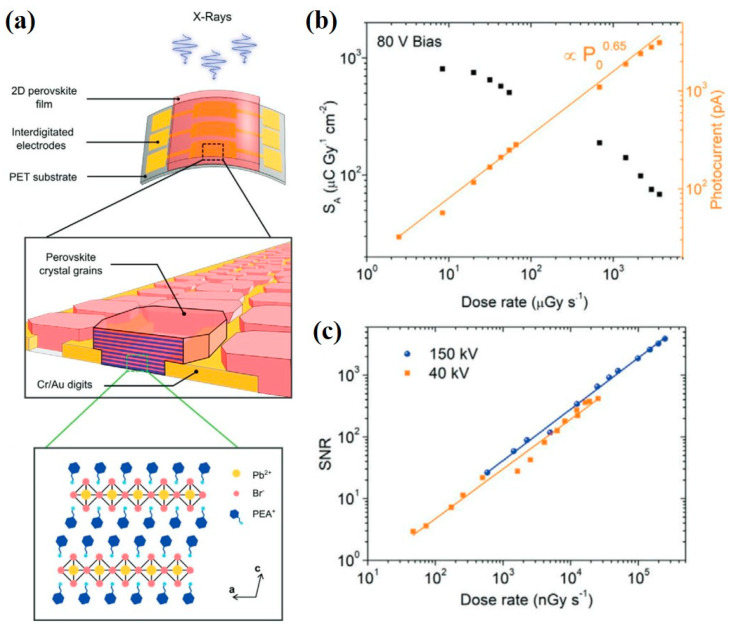
(**a**) Photoconductor-type X-ray detectors by deposition of 2D PEA_2_PbBr_4_ micro-crystalline films on a flexible PET substrate. (**b**) Sensitivity per unit area (S_A_, black curve) and photocurrent (orange curve) as a function of dose rate. (**c**) SNR as a function of dose rate for the device under 150 kVp (blue curve) and 40 kVp (orange curve) accelerating voltages [51].

**Figure 4 nanomaterials-13-02024-f004:**
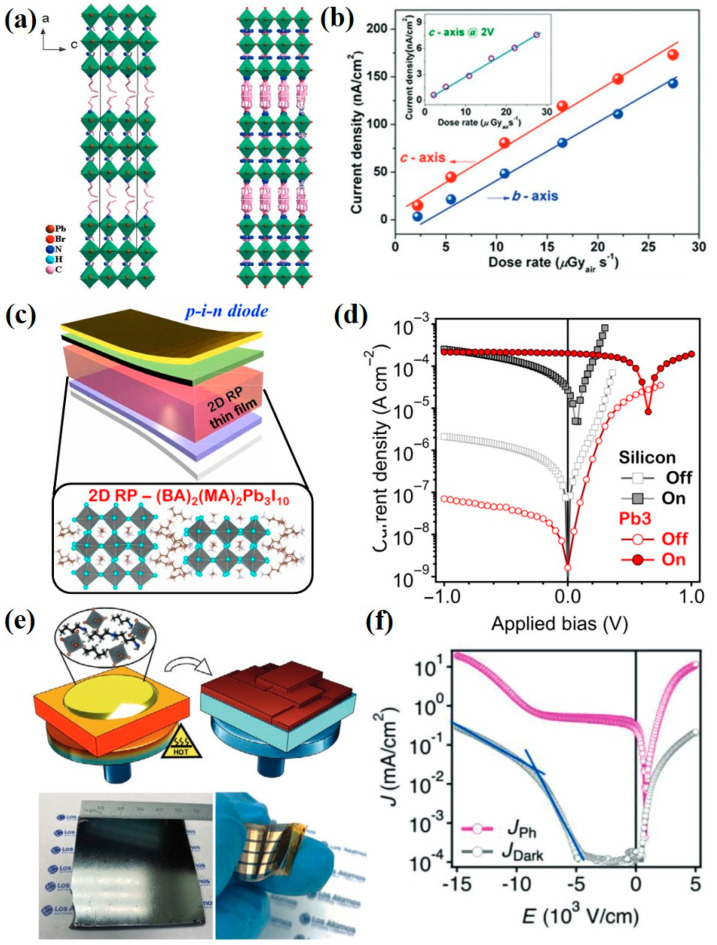
(**a**) Crystal structure of the quasi-2D BA_2_EA_2_Pb_3_Br_10_ perovskite. (**b**) X-ray-generated photocurrent of BA_2_EA_2_Pb_3_Br_10_ perovskite at various dose rates at a bias of 10 V [56]. (**c**) A p–i–n device architecture and (**d**) J–V curve of X-ray detector based on quasi-2D (BA)_2_(MA)_2_Pb_3_I_10_ perovskite [57]. (**e**) Fabrication process of thick film growth of quasi-2D perovskite and (**f**) J–V curve of corresponding X-ray detector [58].

**Figure 5 nanomaterials-13-02024-f005:**
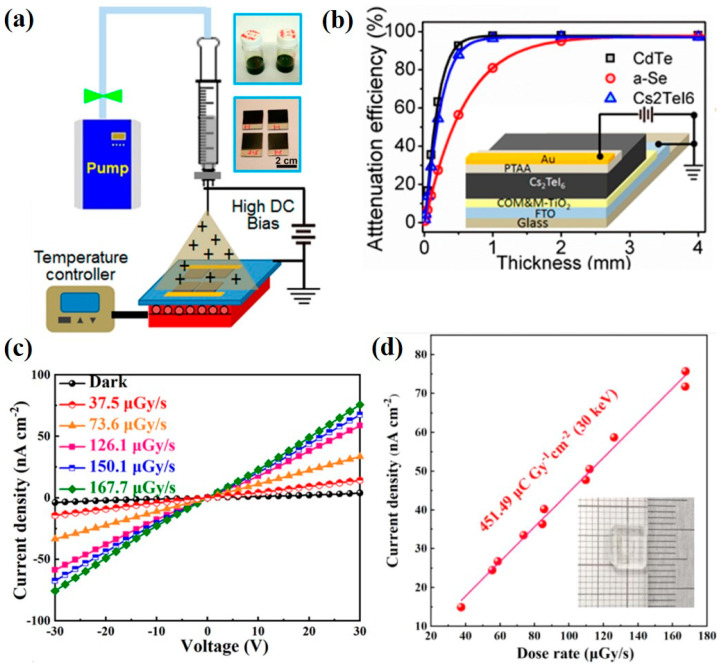
(**a**) Schematic illustration of electrostatic-assisted spray coating process for deposition of large-area Cs_2_TeI_6_ perovskite. (**b**) Comparison of attenuation efficiency of CdTe, a-Se and Cs_2_TeI_6_ with different thicknesses for X-ray irradiation. Inset illustrates the device structure [60]. (**c**) J–V curves and (**d**) photocurrent as a function of dose rate of X-ray detector using Cs_4_PbI_6_ single crystal [61].

**Figure 6 nanomaterials-13-02024-f006:**
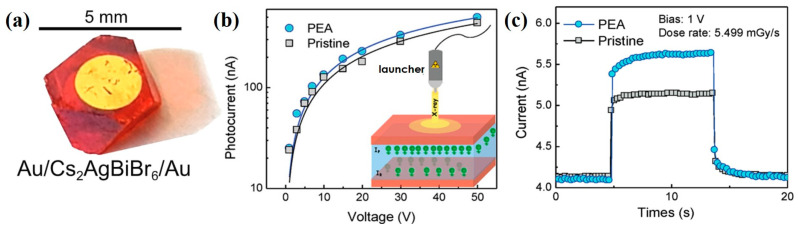
(**a**) Photo of X-ray detector with a device architecture of Au/Cs_2_AgBiBr_6_ perovskite/Au [62]. (**b**) I–V curves of X-ray detectors using pristine Cs_2_AgBiBr_6_ and PEA-Cs_2_AgBiBr_6_ single-crystal. (**c**) X-ray response of the Cs_2_AgBiBr_6_- and PEA-Cs_2_AgBiBr_6_-based devices [64].

**Figure 7 nanomaterials-13-02024-f007:**
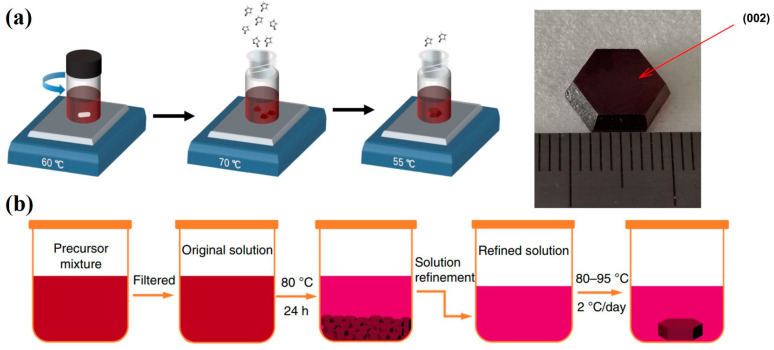
(**a**) FA_3_Bi_2_I_9_ single crystal synthesized by the solution method through controlling the growth mechanism [67]. (**b**) Schematic diagram of preparation of the Cs_3_Bi_2_I_9_ single crystals by the nucleation-controlled solution method [68].

**Figure 8 nanomaterials-13-02024-f008:**
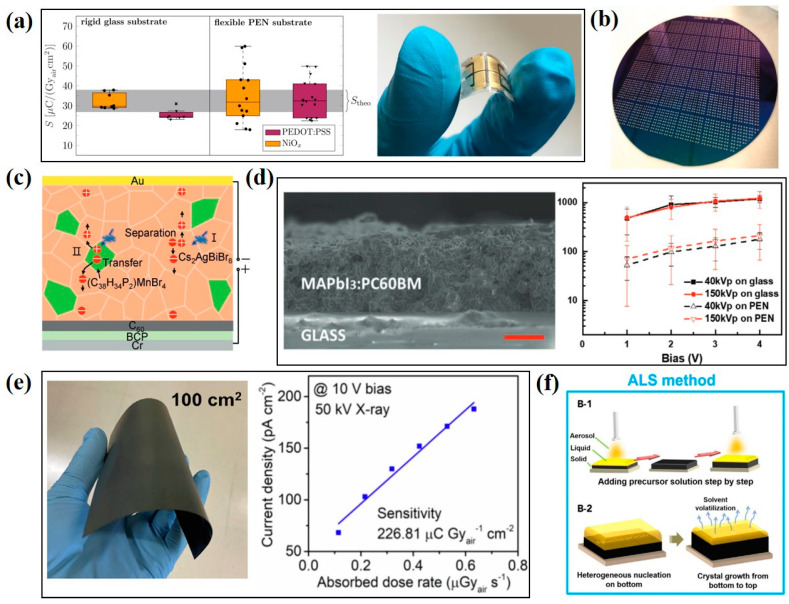
Application of large-area perovskite film for X-ray detectors. (**a**) Reliability of sensitivity for X-ray detectors fabricated on rigid and flexible substrates [69]. (**b**) Photo of large-area CsPbBr_3_ quantum dots deposited on TFT array [70]. (**c**) Illustration of working principle of X-ray detector using mixed Cs_2_AgBiBr_6_/(C_36_H_34_P_2_)MnBr_4_ scintillator [71]. (**d**) Cross-sectional SEM and X-ray photo-response of the device with bar coating of MAPbI_3_:PC60BM mixed absorber [72]. (**e**) Large-area lead-free Cs_2_TeI_6_ perovskite fabricated by electro-spraying on flexible substrates and the relationship between the photocurrent density of Cs_2_TeI_6_-based device and the exposure dose rate [73]. (**f**) Schematic illustration of the perovskite layer prepared by the ALS process [74].

**Table 1 nanomaterials-13-02024-t001:** Summary of key parameters of perovskite-based X-ray detectors.

Crystal Structure	Materials	Crystal Type	Growth Method	Thickness (mm)	E (V·mm^−1^)	uτ (cm^2^·V^−1^)	S (μC·Gy_air_^−1^·cm^−2^)	LoD (nGy_air_·s^−1^)	Ref.
ABX_3_ (Organic)	MAPbI_3_	Single crystals	ITC	NA	100	NA	700,000	1.5	[31]
	MAPbI_3_	Single crystals	ITC	NA	10	1.6 × 10^−3^	NA	NA	[32]
	MAPbI_3_	Single crystals	ITC	1.2 ± 0.04	NA	5.3 × 10^−3^	3.67 × 10^3^	80.6	[33]
	DMAMAPbI_3_	Single crystals	ITC	1.2 ± 0.04	NA	7.2 × 10^−3^	1.18 × 10^4^	16.9	[33]
	GAMAPbI_3_	Single crystals	ITC	1.2 ± 0.04	NA	1.3 × 10^−2^	2.31 × 10^4^	16.9	[33]
	MAPbI_3_	Single crystals	Solution	1	NA	1.49 × 10^−3^	968.9	NA	[34]
	MAPbI_3_	Single crystals	Solution	2~3	3.3	2.57 × 10^−3^	1471.1	46,000	[35]
	MAPbI_3_	Polycrystals (wafer)	Sintering process	0.2~1	200	2 × 10^−4^	2527	NA	[36]
	MAPbBr_3_	Single crystals	ITC	NA	0.83	4.1 × 10^−2^	259.9	NA	[37]
	MAPbBr_3_	Single crystals	Solution	NA	1.43 × 10^4^	NA	359	22,100	[38]
	MAPbBr_3_	Single crystals	Fully textile	0.05	17	NA	12.2 ± 0.6	3000 (for stacked)/8000 (for planar)	[43]
	Bi^3+^-doped MAPbBr_3_	Single crystals	Solution	1.68	31.5	4.12 × 10^−4^	1.72 × 10^3^	NA	[39]
	MA_0.6_Cs_0.4_PbBr_3_	Single crystals	Solution	2	NA	4.64 × 10^2^	2017	1200	[40]
	FA_0.85_MA_0.1_Cs_0.05_Pb I_2.55_Br_0.45_	Single crystals	Thermal evaporation	1	−60	NA	(3.5 ± 0.2) × 10^6^	NA	[41]
	Cs_0.05_FA_0.79_MA_0.16_Pb(I_0.8_ Br_0.2_)_3_	Film	Spin coating	4.5 × 10^−4^	200	NA	3.7 ± 0.1	NA	[42]
ABX_3_ (Inorganic)	CsPbBr_3_	Single crystals	Solution	NA	45	−NA	2552	20,900	[44]
	CsPbBr_3_	Single crystals	Solution	1	20	(2.5 ± 0.2) × 10^−3^	1256	NA	[45]
	CsPbBr_3_	Single crystals	LTC	1	NA	NA	4086	NA	[46]
	CsPbBr_3_	Microcrystals	Solution	0.018	0	NA	470	53	[48]
	CsPbBr_3_	Microcrystals	Vitreous enamel	0.1	1.2 × 10^4^	NA	1450	NA	[49]
	CsPbBr_3_	Quasi-monocrystal	Hot pressing	0.24	4.2	1.32 × 10^−2^	55,684	215	[23]
	CsPbI_2_Br	Film	Co-evaporation	0.001	NA	NA	1.2 × 10^6^	25.69	[50]
2D	(F-PEA)_2_PbI_4_	Single crystals	Solution	2	133	5.1 × 10^−4^	3402	23	[53]
	(PMA)_2_PbI_4_	Single crystals	Cooling crystallization	0.9	NA	8.05 × 10^−3^	283	2.13	[54]
	(F-PEA)_3_BiI_6_	Single crystals	Tablet pressing	NA	100	8.3 × 10^−5^	118.6	30	[55]
	PEA_2_PbBr_4_	Film	Spin coating	1.9 ± 0.8 × 10^−3^	500	1.09 ± 0.07 × 10^−5^	806	42	[51]
Q-2D	BA_2_EA_2_Pb_3_Br_10_	Single crystals	Cooling crystallization	NA	20k	7.6 × 10^−3^	6.8 × 10^3^	5500	[56]
	BA_2_EA_2_Pb_3_I_10_	Film	Hot casting	4.70 × 10^−4^	NA	NA	276,000	10,000	[57]
	MAPbI_3_/n-butylamine iodide	Film	Spin coating	2–3 × 10^−3^	NA	NA	1214	NA	[58]
1D	CsPbI_3_	Polycrystals	Solution	NA	NA	3.63 × 10^−3^	2370	59.7	[59]
0D	Cs_2_TeI_6_	Polycrystals	E-spray deposition	0.5	250	5.2 × 10^−5^	19.2	NA	[60]
	Cs_4_PbI_6_	Single crystals	Solution	NA	NA	9.7 × 10^−4^	451.49	90	[61]
A_2_B_2_X_6_	Cs_2_AgBiBr_6_	Single crystals	Solution	2	NA	NA	105	59.7	[62]
	Cs_2_AgBiBr_6_	Single crystals	Solution	2	25	NA	105	59.7	[63]
	Cs_2_AgBiBr_6_	Single crystals	Solution	2.2	22.7	NA	288	NA	[64]
A_3_B_2_X_9_	MA_3_Bi_2_I_9_	Film	Blade coating	50 × 10^−3^	−3000 V cm^−1^	9.7 × 10^−6^	100.16	98.4	[66]
	FA_3_Bi_2_I_9_	Single crystals	SSCE	0.9	NA	NA	598.1	0.2	[67]
	Cs_3_Bi_2_I_9_	Single crystals	Solution	1.2	50	NA	1652.3	130	[68]
Large area process	MAPbI_3_	Nanocrystals	Bar coating	10 × 10^−3^	800	NA	2300	27,000	[72]
	MAPbI_3_	Polycrystals	Doctor blade coating	0.83	10–240	1.5 × 10^−4^	1.1× 10^4^	300	[22]
	CsPbBr_3_	Monocrystal (quantum dot)	Inkjet printing	2 × 10^−5^	0.1 V	NA	1450	<17,200	[70]
	CsPbI_2_Br	Polycrystals	ALS process	0.04	125	1.14	148,000	280	[74]
	Cs_0.1_(FA_0.83_MA_0.17_)_0.9_Pb(Br_0.17_I_0.83_)_3_	Polycrystals	Inkjet printing	3.7 × 10^−3^	27	2.0 × 10^−6^	59.9	12,000	[69]
	Cs_2_TeI_6_	Polycrystals	Electro-spraying	1.5 × 10^−3^	6670	NA	227	115	[73]
	Cs_2_ABiBr_6_/ (C_38_H_34_P_2_)MnBr_4_	Polycrystals	Tablet pressing	1.7	100	8.5 × 10^−5^	114	200	[71]

ITC: inverse temperature crystallization, LTC: low-temperature crystallization, SSCE: secondary solution constant temperature evaporation, ALS: aerosol–liquid–solid.

## Data Availability

Not applicable.

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
