# Peer review of "Perovskite-Based X-ray Detectors"

_nanomaterials, 2023, doi:10.3390/nano13132024_

Round 1
Reviewer 1 Report
The authors have done a great job in reviewing the capabilities and the potential of different halide perovskites (organic and inorganic) with different structures (single crystal, thin films, 2D, 1D and 0D nanostructures, and double perovskites) as X-ray detectors. Moreover, the possibilities for scale-up process technology for fabricating large-area and thick perovskite films towards commercialization and mass production of perovskite-based X-ray detector, are also discussed.
The manuscript is well organized and scientifically sound, and in my opinion deserves publication in the Special Issue "New Horizon in Perovskite Nanocrystals" of Nanomaterials, after some minor formal corrections are performed.
Aim of the paper
Annalise and review the advantages of halide perovskites for its use as X-ray detectors compared to the state of the art detectors based on amorphous selenium.
No original contributions in the field, as it can be deduced from the main text, figure captions (figures are borrowed from previous publications) and summary table (again, data taken from previous publications, no new original data)
Main contributions
Annalise the advantages and drawbacks of the halide perovskite systems attending to its composition, morphology and structure.
Present possible scale processes to move from the lab to the mass production
What the findings are
Using emerging perovskite materials can provide a better X-ray detection performance, a lower cost as well as a less environmental pollution
Polycrystalline-perovskite-based X-ray detectors show a poorer performance than the single crystalline counterpart because of the increased grain boundaries
Organic-inorganic hybrid perovskites suffer of poor environmental stability due to the rapid degradation of perovskites exposed to humidity, heat, light and oxygen
All inorganic perovskite materials, show better environmental stability
The sensitivity of X-ray detectors using 2-D perovskite is generally lower than that of 3-D perovskite counterpart
Application of 1-D needle-like inorganic perovskite, such as CsPbI3, in the X-ray detector shows a high X-ray sensitivity owing to its high resistivity and large carrier mobility–lifetime product.
0-D all-inorganic perovskite material of Cs2TeI6 is demonstrated as a potential candidate for the X-ray detector in virtue of its high X-ray sensitivity and environmental stability
Double perovskites are highly stable and show better attenuation to X-ray due to the introduction of large Z-numbered element.
Environmental-friendly lead-free perovskite exhibits potential application in X-ray detectors in virtue of its high attenuation to X-rays
Large-area fabrication methods for preparing the thick perovskite film, such as inkjet printing, doctor blade coating, spray coating and aerosol-liquid-solid (ALS) processes, can be utilized to fabricate large-area perovskite-based X-ray detectors
Its strengths
In the first part, complete introduction on the relevant parameters that define the efficiency and figure of merit of X-ray detectors.
Exhaustive and detailed review on the work done up to now on the subject towards the use of halide perovskites for X-Ray detectors.
Lines 183 and 184, "when the device is exposure to X-ray exposure" weird sentence ?
Table I caption: "Summary of...." instead of "Summaries of.."
Edit table I, in a way that it reads better. First title line very crowded, with mixing of units etc..
Maybe it is due to the version for review, but in gneral the quality of the figures is poor and the lettering is dificult to read.
Reviewer 2 Report
A detailed review of X-ray detectors designed on the basis of perovskite-like compounds summarises 72 literature sources. The period from 1964 to 2022 is covered. The review is very relevant because of the significant interest in this topic. Various types of X-ray sensors were analysed, and recommendations for further improvement of their properties were offered.
The disadvantages include the low resolution of the graphic material as well as its complexity with numerous panels.
Reviewer 3 Report
The paper provides a through snapshot of the development and application of Perovskite materials to x-ray image detection. The final summary table is particularly useful.
With some careful proof reading it would be acceptable as a publication in Nanomaterials. I have noted a few guides to editing in the comments below. A thorough review of the English would help.
In some parts the cited lists of previous works is rather disordered, and without linking sentences. I suggest either creating a table, or perhaps attempting to link only the major investigations only with text.
In section 1.2 it would be worth pointing out the difference in charge transport between using an intrinsic semiconductor material such as selenium with ohmic contacts, and semiconductor junction system.
Page 6, line 250: The units used for sensitivity should be checked throughout the script. There is an ambiguity as to whether the dose rates refer to those in the material, or with the ‘air’ subscript the incident air KERMA. In one case (sec. 3.2.4, line 476) non-SI units are used.
Some specific typographical and editing comments:
Abstract line 20: ‘detailly’
Section 1.1, line 68: The use of the word ‘facile’ in the script is not appropriate.
Page 4, line 170 ‘trap trapping’
Section 2.2.4 line 214: The mu-T product is a characteristic of the x-ray converter material, not the x-ray detector.
Section 2.2.4, line 217: No need to reproduce equation 8 here.
Section 3, line 233: Alternatives to ‘excellence’ and ‘compatible to’
Section 3.1.1.1: Use of un-scientific terms such as ‘superb’ and ‘great’ should be avoided.
Section 3.2, line 387: ‘n increasing continuously to infinity’ is an abstract concept, out of place in this discussion.
Page 17, line 643 – Avoid using ‘1-D’ to describe the needle like structures and explain what is meant by ‘0-D’ materials here.
Page 17, line 661: Expand the acronym ALS.
The figures panels are quite small when viewed on A4 pages. If this is necessary to keep the overall page count down. Figures 1, and 4 sections e) might be deleted to allow an increase in the overall figure size.
In general the English is acceptable. There is some use of inappropriate words, and the flow could be improved.
